# Cytosolic HMGB1 Mediates LPS-Induced Autophagy in Microglia by Interacting with NOD2 and Suppresses Its Proinflammatory Function

**DOI:** 10.3390/cells11152410

**Published:** 2022-08-04

**Authors:** Seung-Woo Kim, Sang-A Oh, Song-I Seol, Dashdulam Davaanyam, Ja-Kyeong Lee

**Affiliations:** 1Department of Biomedical Sciences, Inha University School of Medicine, Inchon 22212, Korea; 2Department of Anatomy, Inha University School of Medicine, Incheon 22212, Korea

**Keywords:** HMGB1, NOD2, autophagy, inflammation, microglia

## Abstract

The high mobility group box 1 (HMGB1), a well-known danger-associated molecule pattern (DAMP) molecule, is a non-histone chromosomal protein localized in the nucleus under normal physiological conditions. HMGB1 exhibits diverse functions depending on its subcellular location. In the present study, we investigated the role of HMGB1-induced autophagy in the lipopolysaccharide (LPS)-treated BV2 microglial cell line in mediating the transition between the inflammatory and autophagic function of the nucleotide-binding oligomerization domain-containing 2 (NOD2), a cytoplasmic pattern-recognition receptor. The induction of the microtubule-associated protein 1 light chain 3 (LC3), an autophagy biomarker, was detected slowly in BV2 cells after the LPS treatment, and peak induction was detected at 12 h. Under these conditions, NOD2 level was significantly increased and the binding between HMGB1 and NOD2 and between HMGB1 and ATG16L1 was markedly enhanced and the temporal profiles of the LC3II induction and HMGB1-NOD2 and HMGB1-ATG16L1 complex formation coincided with the cytosolic accumulation of HMGB1. The LPS-mediated autophagy induction was significantly suppressed in BV2 cells after HMGB1 or NOD2 knock-down (KD), indicating that HMGB1 contributes to NOD2-mediated autophagy induction in microglia. Moreover, NOD2-RIP2 interaction-mediated pro-inflammatory cytokine induction and NF-κB activity were significantly enhanced in BV2 cells after HMGB1 KD, indicating that HMGB1 plays a critical role in the modulation of NOD2 function between pro-inflammation and pro-autophagy in microglia. The effects of the cell-autonomous pro-autophagic pathway operated by cytoplasmic HMGB1 may be beneficial, whereas those from the paracrine pro-inflammatory pathway executed by extracellularly secreted HMGB1 can be detrimental. Thus, the overall functional significance of HMGB1-induced autophagy is different, depending on its temporal activity.

## 1. Introduction

The high mobility group box 1 (HMGB1) is a well-known damage-associated molecular pattern (DAMP) molecule. However, depending on its subcellular location, HMGB1 exhibits diverse functions. As a nuclear protein, HMGB1 is involved in DNA replication and repair, transcription regulation, and chromatin structure-modulation as a non-histone chromosomal protein [1,2,3]. When it is present in the extracellular milieu after being released from dying cells or secreted from activated immune cells, HMGB1 induces and aggravates inflammatory responses [4,5]. Interestingly, cytosolic HMGB1 plays a critical role as an autophagy regulator; starvation-induced oxidative stress promotes the translocation of HMGB1 from the nucleus to the cytoplasm and the translocated HMGB1 enhances the autophagic flux via direct interaction with Beclin-1 [6]. Controlling the switch between the pro-autophagic and pro-apoptotic functions of Beclin 1 and autophagy-related 5 (ATG5) by cytosolic HMGB1 during inflammation has also been reported in inflammatory bowel disease [7]. In addition, HMGB1-p53 complex regulates the tumor cell survival and death by promoting either apoptosis or autophagy through the modulation of the cytoplasmic translocation of the reciprocal binding partner [8].

The nucleotide-binding oligomerization domain (NOD) acts as a cytoplasmic pathogen recognition sensor, functioning as a host defense against pathogens and inflammation [9,10]. NOD1 and NOD2 recognize the peptidoglycans (PGNs) localized on the bacterial cell wall; in particular, NOD2 recognizes bacteria by interacting with muramyl dipeptide (MDP), a PGN present both in Gram-positive and Gram-negative bacteria, through its C-terminal leucine-rich repeats domain [11,12]. The NOD1 and NOD2 activation initiates the pro-inflammatory processes by activating the receptor-interacting protein 2 kinase (RIP2, CARD-containing serine/threonine kinase), and subsequently the activation of nuclear factor-kappa B (NF-κB) [13]. Interestingly, MDP (a NOD agonist) also increases the number of autophagosomes in bone marrow-derived macrophages, and NOD1 and NOD2 play a crucial role in autophagy initiation in response to invasive bacteria via the recruitment of autophagy-related protein 16-1 (ATG16L1) [14]. In addition, ATG16L1 regulates the NOD-induced inflammatory response by interfering with the poly-ubiquitination of the RIP2 adaptor and recruitment of RIP2, resulting in anti-inflammatory function [15]. Therefore, NOD2 plays an important role in inducing inflammation and autophagy, and these two functions seem to be closely associated.

The extracellular secretion of HMGB1 occurs in diverse ways. Post-translational modifications regulate the secretion of HMGB1 [16] and stimuli triggering lysosomal exocytosis regulate the extracellular secretion of HMGB1 [17]. The extracellular delivery of interleukin (IL)-1β and HMGB1 via an autophagy-based unconventional secretory pathway has been reported, and critical roles for ATG5, inflammasomes, and one of the two mammalian Golgi reassembly-stacking protein (GRASP) paralogs, GRASP55 (GORASP2), have also been shown [18]. The cytosolic HMGB1 increases the autophagosome formation, which subsequently induces the extracellular secretion of HMGB1 under various pathological conditions [7,19,20,21]. HMGB1 plays a vital role in various pathological conditions in the central nervous system (CNS), and numerous studies have reported the activation of microglia and aggravation of its pro-inflammatory function by HMGB1. We have previously reported that the LPS treatment induces the HMGB1 translocation from the nucleus to the cytosol and to the extracellular environment in the ischemic brain and microglia [5,22,23,24]. In the present study, we investigated the putative role of cytosolic HMGB1 in autophagy induction, in particular, whether cytosolic HMGB1 modulates the function of NOD2 between inflammation and autophagy by interacting with NOD2 using the LPS-treated BV2 microglial cell line.

## 2. Materials and Methods

### 2.1. Cell Cultures and Treatment

The BV2 murine microglial cell line was maintained in Dulbecco’s modified Eagle’s medium (DMEM; Sigma, St. Louis, MO, USA), supplemented with 5% heat-inactivated fetal bovine serum (Hyclone, Logan, UT, USA), penicillin (20 U/mL), and streptomycin (20 mg/mL) in a 5% CO_2_ incubator. The BV2 cells were stimulated with lipopolysaccharide (LPS, obtained from *Escherichia coli* O111: B4) at the concentration as indicated in each experiment. For the inhibitor treatment, the BV2 cells were pretreated with Bafilomycin A1 (BA; 50 nM, Sigma, St. Louis, MO, USA), Chloroquine (CQ; 5 μM, Sigma), or 3-methyladenine (3MA; 5 mM. Caymanchem, Ann Arbor, MI, USA) for 2 h and then treated with LPS (1 μg/mL) for 24 h.

### 2.2. siRNA Transfection

The siNOD2 and siCon (control siRNA) were purchased from Integrated DNA Technologies, Inc. (Coralville, IA, USA). The siNOD2 and siCon (80 pmol each) were individually mixed with Oligofectamine and Opti-MEM (both Invitrogen, Carlsbad, CA, USA), according to the manufacturer’s instructions, and then the siRNA-Oligofectamine complex was added to the BV2 cells (6 × 10^4^ cells/well) and incubated for 24 h.

### 2.3. Generation of HMGB1 Knockdown (KD) Stable Cell Line

To generate the HMGB1 knockdown BV2 cells, a stable transfection of the plasmid-expressing HMGB1 shRNA (5′-GATCCCGAAGCACCCGGATGCTTCTTTCAAGAGAAGAAGCATCCGGGTGCTTCTTTTTTGGAAA-3′) was conducted [5]. The HMGB1 shRNAs were synthesized and subcloned into the pU6 plasmid (Ambion, Austin, TX, USA), and then transfected into BV2 cells [5]. The cells with reduced HMGB1 expression were selected, and stable cell lines were established by selection with 400 µg/mL G418 for six weeks. The level of HMGB1 was analyzed using immunoblot analysis, and a stable cell line showing the lowest HMGB1 level was selected shHMGB1-BV2 (3-5).

### 2.4. Nuclear Extract Preparation

The nuclear extracts were prepared from BV2 cells (5 × 10^6^ cells) using nuclear extraction kits (Thermo Fisher Scientific, Rockford, IL, USA), according to the manufacturer’s instructions. The crude nuclear proteins in the supernatants were collected and stored at −70 °C until further use.

### 2.5. Cell Culture Media Preparation

The culture medium was collected and concentrated using a Nanosep^®^ centrifugal device (Pall Life Science, Port Washington, NY, USA). Briefly, 500 μL of culture medium was harvested in the upper reservoir of the device and centrifuged for 10 min at 14,000 rpm at room temperature. The concentrated medium was collected and stored at −70 °C until further use.

### 2.6. Immunofluorescence Staining

The BV2 cells were washed once with ice-cold phosphate-buffered saline (PBS) and fixed with 4% paraformaldehyde (Sigma Aldrich, St. Louis, MO, USA) for 20 min. Fixed BV2 cells were blocked and permeabilized with 1% goat serum and 0.1% Triton X-100 in PBS, followed by incubation with primary antibodies for anti-light chain 3 (LC3; 1:200; PM036, MBL, Woburn, MA, USA) and anti-HMGB1 1:200; ab190377, Abcam, Cambridge, UK) at 4 °C overnight. For the double immunostaining, FITC ant-rabbit IgG (Merck Millipore, Burlington, MA, USA) or rhodamine anti-mouse IgG (Merck Millipore) was used as the secondary antibody and incubated for 1 h at room temperature. The cells were counterstained with DAPI (Merck Millipore) to visualize the nuclei and then observed under a fluorescence microscope (Axio Observer; Zeiss, Oberkochen, Germany).

### 2.7. Immunoblotting

The BV2 cells were washed twice and collected in ice-cold PBS. The cells were resuspended at 4 °C in RIPA buffer containing 150 mM NaCl, 50 mM Tris-HCl (pH7.4), 0.5% Triton X-100, 0.5% NP-40, 0.25% sodium deoxycholate, and complete mini protease-inhibitor cocktail tablets (Roche, Basel, Switzerland), and incubated on ice for 15 min. The samples were centrifuged for 10 min at 14,000 rpm at 4 °C, and the supernatant was harvested. Equal amounts of protein were loaded onto 6–12% sodium dodecyl sulfate-polyacrylamide gel electrophoresis (SDS-PAGE) gels and transferred to polyvinylidene fluoride (PVDF) membranes (Merck Millipore, Darmstadt, Germany), and blocked with 5% non-fat dry milk in Tris-buffered saline (25 mM Tris, 2.7 mM KCl, 137 mM NaCl, and 0.1% Tween-20; pH 7.4). The complete experimental procedure is previously described [25]. In brief, after blocking, the membrane was incubated with target primary antibodies overnight in a cold room at 4 °C with constant shaking. The primary antibodies for anti-LC3 (1:2000; 2775S, Cell Signaling Technology, Danvers, MA, USA), anti-HMGB1 (1:2000; ab18256, Abcam, Cambridge, UK), anti-alpha tubulin (1:10,000; GTX112141, Genetex, Irvine, CA, USA), anti-LaminB1 (1:2000; D9V6H, Cell Signaling Technology), anti-ATG5 (1:1000; SC-33210, Santa Cruz, Dallas, TX, USA), anti-ATG7 (1:2000; D12B11, Cell Signaling Technology), anti-ATG14 (1:2000; 28021-1-AP, Proteintech, Chicago, IL, USA), anti-ATG16 (1:2000; D6D5, Cell Signaling Technology), anti-Beclin-1 (1:5000; 11306-1-AP, Proteintech), anti-SQSTM1/p62 (1:2000; ab56416, Abcam), and anti-NOD2 (1:1000, DF12125, Affinity Biosciences, Cincinnati, OH, USA) were diluted as indicated for each antibody. The membrane was washed with Tris-buffered saline with 0.1% Tween (TBST) and incubated with target secondary antibody for 1 h. The membrane was washed in TBST twice for 10 min each and the signals were detected, using a chemiluminescence kit (Merck Millipore). The signals were confirmed by band intensity analysis, using Image J software.

### 2.8. Immunoprecipitation-Linked Immunoblot Analysis

The cell lysates (0.5–1 mg of total protein) were immunoprecipitated with anti-HMGB1 (Abcam), or anti-NOD2 (Santa Cruz) at 4 °C overnight on protein A-Sepharose beads. The bead-bound complexes were pelleted, washed several times with lysis buffer, and boiled with a SDS sample buffer for 5 min prior to SDS-PAGE. For the Western blot analysis, the proteins were transferred to PVDF membranes after SDS-PAGE and blocked with 5% non-fat dry milk. The blots were incubated with specific primary antibodies, such as anti-HMGB1 (1:2000; ab190377, Abcam,), anti-NOD2 (1:1000; DF12125, Affinity Bioscience), or anti-ATG16 (1:2000; 8089, Cell Signaling Technology), overnight at 4 °C. The signals were detected, using a chemiluminescence kit (Merck Millipore).

### 2.9. RNA Preparation and Reverse Transcription-Polymerase Chain Reaction (RT-PCR)

The cell lysates were collected from BV2 and shHMGB1-BV2 cells at 4, 6, 8, and 12 h after the LPS treatment and the total RNA was prepared using TRIzol reagent (Thermo Fisher Scientific) and 1 µg RNA samples were used for cDNA synthesis using a RT-PCR kit (Roche, Mannheim, Germany), following the manufacturer’s instructions. The sequences of the primer sets are summarized in Table 1.

### 2.10. Statistical Analyses

The statistical analyses were performed using analysis of variance (ANOVA) followed by the Newman–Keuls test. All of the results are presented as the mean ± SEM. The statistical significance was set at (*p* < 0.05).

## 3. Results

### 3.1. Temporal Profile of Autophagy Induction in LPS-Treated Microglia

When we examined the temporal profile of autophagy induction in LPS (1 μg/mL)-treated microglia, the induction of the microtubule-associated protein 1 (LC3), a biomarker of autophagy, occurred slowly and the peak induction was detected at 12 h post treatment and a significantly enhanced level was sustained at 24 h (Figure 1A,B). In contrast, the protein level of the SQSTM1/p62 (sequestome 1) gradually decreased (Figure 1A). When we stained the BV2 cells with anti-LC3 antibody, the immunoreactivity of LC3 was redistributed after the LPS treatment from a diffuse pattern to the typical punctate type (Figure 1C). The number of LC3 puncta was significantly increased, reaching a peak at 12 h after the LPS treatment and a significantly enhanced level was sustained at 24 h (Figure 1D). In addition, the expression of the other autophagy-related molecules, such as ATG7, ATG14, and ATG16, was upregulated in the LPS-treated BV2 cells and the maximum induction detected at 12 h or 24 h after the LPS treatment (Figure 1E,F). The expression of Beclin-1, a well-known key regulator of autophagy, was also significantly induced and the maximum induction was also detected at 12 h after the LPS treatment (Figure 1E,F). When we disrupted the autophagy flux by pre-treating the BV2 cells with Bafilomycin A1 (Baf) or chloroquine (CQ), which inhibits autophagosome-lysosome fusion, the LC3II level at 24 h after the LPS treatment was significantly increased (Figure 1G,H), indicating that the autophagy flux was impaired. The LC3 puncta accumulated in the cytosol of the BV2 cells and their number significantly increased at 12 h and 24 h after the LPS treatment (Appendix A). It is important to note that the autophagy induction in the LPS-treated BV2 cells was significantly suppressed after treatment with a Toll-like receptor 4 (TLR4) antagonist, TLR4-IN-C34 (Figure 1I,J), indicating a critical role for TLR4 signaling. Collectively, these results indicate that autophagy was slowly induced in a TLR4-dependent manner in the LPS-treated BV2 cells, that it peaked at 12 h, and then decreased thereafter.

### 3.2. NOD2 Plays a Critical Role in LPS-Induced Autophagy in BV2 Cells

To investigate whether NOD2 is involved in autophagy induction in microglia after the LPS treatment, the NOD2 level was examined in the BV2 cells after treatment with LPS (500 ng/mL). The amount of NOD2 was significantly increased after the LPS treatment, reaching the maximum level at 12 h (Figure 2A). To further examine the importance of NOD2, the LC3-II induction was examined in the BV2 cells wherein the NOD2 expression was knocked down (KD) via transfection with NOD2 siRNA (siNOD2-BV2). The basal NOD2 level was reduced to 13.2 ± 7.6% of that in the control cells and this reduction was not detected in the cells transfected with the scrambled siRNA of NOD2 (siSC-BV2) (Figure 2B). Notably, the LC3II induction observed at 12 h after the LPS treatment (Figure 1A,B) was not detected in the siNOD2-BV2 cells, where the basal level of LC3II was 29.9 ± 9.1% of that in the control cells and the LPS-mediated induction was not detected (Figure 2C). However, such a basal level reduction in LC3II was not observed and its induction after the LPS treatment was detected in the siSC-BV2 cells (Figure 2C), indicating that NOD2 plays an important role in the autophagy induction in the LPS-treated BV2 cells. In addition, the basal level of the autophagy-related molecules (ATG7, ATG5, and ATG16) was also significantly suppressed in the NOD2 KD BV2 cells and induction after the LPS treatment was not observed (Figure 2D,E). These results further support a notion that NOD2 plays a critical role in the LPS-mediated induction of autophagy in microglia.

### 3.3. Cytoplasmic HMGB1 Binds to NOD2 and ATG16 in LPS-Treated BV2 Cells

To examine the role of the cytosolic HMGB1 in the autophagy induction in the LPS-treated BV2 cells and its association with NOD2, we investigated the temporal profile of the HMGB1 translocation from the nucleus to the cytoplasm and its accumulation in the culture media in the LPS-treated BV2 cells. We found that HMGB1 remained in the cytoplasm for more than 12 h (between 6 and 18 h) before it was secreted extracellularly (Figure 3A–C). Interestingly, the immunofluorescence staining of the BV2 cells with anti-HMGB1 at 18 h after the LPS treatment revealed that the cytosolic HMGB1 was detected as diffusely stained puncta (Figure 3D), indicating the participation of HMGB1 in autophagosome formation. These observations prompted us to examine whether HMGB1 interacts directly with NOD2 in the process of autophagosome formation. The co-immunoprecipitation analysis revealed an enhanced interaction between HMGB1 and NOD2 first at 6 h after the LPS treatment, it continued to increase, and peaked at 12 h (Figure 3E). We also found that HMGB1 interacts with ATG16L1 at 6 h after the LPS treatment, and this interaction was maintained until 12 h and then decreased (Figure 3F). Notably, the time points when HMGB1 was co-immunoprecipitated with NOD2 or ATG16L1 coincided with the period of HMGB1 localization in the cytoplasm (Figure 3A,B). These results indicate that the cytosolic HMGB1 interacts with NOD2 and ATG16 and may be involved in the autophagosomal complex formation.

### 3.4. Suppression of LPS-Induced Autophagy in HMGB1 KD BV2 Cells

To further investigate the importance of HMGB1 in autophagy formation in the LPS-treated BV2 cells, we generated HMGB1 KD BV2 cell lines by stably transfecting shHMGB1 (shHMGB1-BV2). Among the few lines showing stable transfection, the shHMGB1-BV2 (3-5) cell line was selected and used (Appendix A). In the shHMGB1-BV2 (3-5) cells, the HMGB1 level was reduced to 20.7 ± 11.9% of that in the wild type, and the ratio between the levels in the nucleus and cytoplasm seemed to be maintained (Figure 4A,B). Importantly, the basal LC3 level in the shHMGB1-BV2 cells was significantly higher than that in the wild type; however, the LPS-mediated LC3 induction was not detected in the shHMGB1-BV2 cells at all of the examined time points, including the peak induction at 12 h (Figure 4C). This observation indicates an important role for HMGB1 in the LPS-induced microglial autophagy. In addition, the significant reduction in the autophagosome-like dots at 12 h post-LPS treatment visualized using anti-LC3 staining further strengthened the critical role of HMGB1 in this process (Figure 4D,E). Interestingly, the basal level of NOD2, ATG16, and Beclin-1 was significantly higher in the shHMGB1-BV2 cells compared to the normal BV2 cells (Figure 4F–I). However, further induction after the LPS treatment was not observed for ATG16, and Beclin-1; in contrast, it was observed for NOD2 (Figure 4F–I). Together, these results indicate that HMGB1 plays an important role in autophagy in the LPS-treated microglia.

### 3.5. HMGB1 KD Augments Inflammation in LPS-Treated BV2 Cells at Early Time Points

The significant suppression of the LPS-induced LC3 upregulation and puncta formation, with a significant induction of the NOD2 basal level in the shHMGB1-BV2 cells prompted us to examine if the pro-inflammatory cytokine induction was enhanced in the shHMGB1-BV2 cells. In the normal BV2 cells, inducible nitric oxide synthase (iNOS) induction gradually increased between 4 h and 12 h after the LPS treatment (Figure 5A,B). Interestingly, in the BV2-shHMGB1 cells, the iNOS induction at 4 h after the LPS treatment was significantly higher and this enhanced induction was sustained at 6 h after the LPS treatment (Figure 5A,B). Importantly, similarly enhanced inductions of IL-1β, IL-6, and tumor necrosis factor (TNF) α were observed in the BV2-shHMGB1 cells with some variations in the differences in induction levels and time points (Figure 5A,C–E). The results indicate an enhanced pro-inflammatory response during this early period after the LPS treatment in the HMGB1 KD BV2 cells.

### 3.6. Suppression of Autophagy Augments Proinflammatory Response in LPS-Treated BV2 Cells at Early Time Points

Since the basal level of NOD2 is significantly higher in the HMGB1 KD BV2 cells than that in the normal BV2 cells (Figure 4F,G), we investigated the possibility that this enhanced NOD2 is involved in the augmentation of pro-inflammation but not the pro-autophagic function, due to the limited cytosolic HMGB1 in the HMGB1 KD BV2 cells. To investigate this possibility, we examined the pro-inflammatory cytokine expression after inhibiting autophagy by pre-treating the normal BV2 and HMGB1 KD BV2 cell lines with 3-MA (5 mM), an autophagy inhibitor, for 2 h. Interestingly, the levels of the pro-inflammatory cytokine (iNOS, IL-1β, and IL-6) at 4 h after the LPS treatment were significantly higher in the wild-type-BV2 cells (Figure 6A–D). In contrast, in the BV2-shHMGB1 cells, pro-inflammatory cytokine expression at 4 h after the LPS treatment was higher than those in the normal BV2 cells, as observed in Figure 5, but no additional increase was detected after the 3-MA treatment (Figure 6A–D), suggesting that the enhanced pro-inflammation in the HMGB1 KD BV2 cells may result from the suppressed autophagy. Together these results support that the cytosolic HMGB1 may be involved in the modulation of the pro-inflammatory and pro-autophagic function of NOD2 in the LPS-treated BV2 cells, and it was tilted to a pro-inflammatory function in the HMGB1 KD condition.

## 4. Discussion

The present study demonstrates that the cytosolic HMGB1 plays a critical role in the LPS-induced autophagy in BV2 cells. We showed that, under the LPS treatment, the cytosolic HMGB1 that translocated from the nucleus interacts with NOD2, promoting the formation of autophagosomes. In addition to Beclin-1 and p53, whose interaction with HMGB1 modulates the pro-apoptotic versus pro-autophagic fate of the target cell [6,7,8], here, we added NOD2 as a novel target whose functions between pro-inflammation and pro-autophagy are modulated by HMGB1 (Figure 6E). To the best of our knowledge, this is the first report showing a direct interaction between HMGB1 and NOD2 and the novel HMGB1 function in modulating NOD2 during infection.

NOD2 is an intracellular pattern-recognition receptor for MDP. The NOD2 activation is involved in autophagy induction during Crohn’s disease pathogenesis [14,26] and in pneumococcal meningitis [27]. Here, we showed that NOD2 is also activated under the LPS treatment and interacts with HMGB1 and probably together with ATG16L1, and that this complex may contribute to autophagosome formation. Moreover, the suppression of the LPS-mediated autophagy induction in the NOD2 KD BV2 cells further substantiates the vital role of NOD2 in autophagy induction. In addition to NOD2, we showed a critical role for TLR4 in the LPS-induced autophagy, as has been reported in the primary human macrophages and murine macrophage cell lines by Xu et al. (2007) [28]. TLR4 exerts a dual function during the LPS-induced autophagy initiation in BV2 cells: microglia activation and then the HMGB1 translocation from the nucleus to the cytoplasm [29], and the subsequent NOD2 activation (Figure 6E). Therefore, two pattern-recognition receptors, namely, TLR4 and NOD2, are involved in the LPS-induced autophagy in BV2 cells. Notably, the synergistic effect of NOD2 and TLR4 on the activation of autophagy has been reported in human submandibular gland inflammation [30], wherein NOD2 and TLR4 were activated by their specific agonists, MDP and LPS, respectively.

In the present study, we found that the temporal profiles of LC3II induction and HMGB1-NOD2 or HMGB1-ATG16L1 complex formation coincided with the cytosolic accumulation of HMGB1. Upon moving to the extracellular space, HMGB1 remains in the cytosol for longer than 12 h in BV2 cells (Figure 3A), and during this period, cytosolic HMGB1 may function as a ligand for the intracellular NOD2 receptor and shift its function to pro-autophagy. It is interesting to note here that a similar modulation of NOD2 signaling by extracellular HMGB1, not cytosolic HMGB1, has been reported in an animal model of acute lung injury, wherein extracellular HMGB1/TLR4 signaling activates NOD2, which induces autophagy and suppresses NOD2-RIP2 signaling [31]. Therefore, in addition to Beclin-1 and p53, we established NOD2 as a novel target for HMGB1, whose function is modulated by cytosolic HMGB1.

It is intriguing to note that the basal LC3II level was significantly higher in the shHMGB1-BV2 cells than in the normal BV2 cells and, similarly, the basal levels of NOD2, ATG16L1, and Beclin1 were also significantly enhanced (Figure 4). However, no additional induction of LC3II or other autophagy-related molecules occurred after the LPS treatment (Figure 4). We cannot explain why the basal level of autophagy (LC3II induction and puncta formation) and autophagy-related molecule expression were increased in the shHMGB1-BV2 cells, wherein the level of HMGB1, a cargo for autophagosomes, was low. This may be because slightly enhanced autophagosomes are necessary to secrete pro-inflammatory cytokines, such as IL-1β, whose production was significantly increased early after the LPS treatment in the shHMGB1-BV2 cells, as shown in Figure 5. It is also possible that other basal autophagic pathways may be activated in the absence of HMGB1. However, further studies regarding the relevance of this phenomenon and underlying molecular mechanism are required.

Emerging evidence indicates that autophagy may be a core regulator of CNS aging and neurodegeneration. In particular, microglial autophagy plays a vital role in maintaining brain homeostasis and the modulation of neuroinflammation. The neuroprotective effects of microglial autophagy have also been reported in acute CNS pathologies, such as traumatic brain injury [32] and cerebral ischemia [33,34], and impairing microglial autophagy aggravates the pathologies of Alzheimer’s disease [35,36] and Parkinson’s disease [37]. In addition, deficient autophagy in microglia impairs the synaptic pruning, causes social and behavioral defects [38], and aggravates depression-like behavior [39,40]. Recently, the interrelationship between autophagy induction and M1/M2 transition of macrophages or microglia has been reported in atherosclerosis [41] and brain infection [42], in which autophagy promotes the cell fate to M2 and M1, respectively. Therefore, although overall the microglial autophagy seems beneficial both in the physiology and pathology of the brain, there are also a few examples of the detrimental effects of autophagy [43,44]. However, the consequences of autophagy induction by cytosolic HMGB1 and its secretion by microglia in vivo might be complicated depending on the time point due to the interplay between different brain cells. Here, we demonstrate that it is beneficial for the microglia, whose fate changes from pro-inflammation to pro-autophagy; however, it may be detrimental to neighboring cells, such as neurons, astrocytes, and endothelial cells, since the extracellular HMGB1 secreted from the microglia via autophagosomes exerts pro-inflammatory effects on them. Therefore, the effects of the autocrine pathway by cytosolic HMGB1 occurring in the microglia and those from the paracrine pathway by extracellular HMGB1 are different, depending on the temporal progress (Figure 6E).

The present study reveals that cytosolic HMGB1 is involved in autophagic flux in the LPS-treated BV2 cells by interacting with NOD2, which modulates inflammation and autophagy. Further studies are needed to determine how HMGB1 regulates the autophagic process, specifically the interaction with various factors in diverse cellular stresses or diseases, and how it modulates its secretion based on these findings.

## Figures and Tables

**Figure 1 cells-11-02410-f001:**
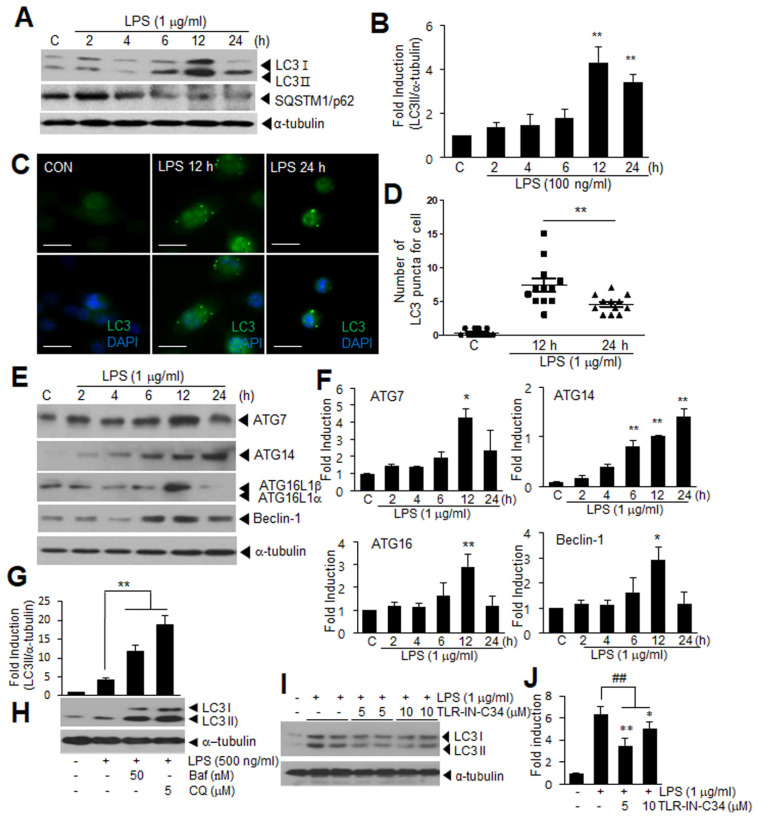
LPS induces autophagy in BV2 cells. BV2 cells were treated with LPS (1 μg/mL) for 2, 4, 6, 12, and 24 h (**A**–**F**). (**A**,**B**) Protein levels of LC3 and SQSTM1/p62 were examined using immunoblotting. Representative immunoblots are presented in (**A**), and quantification results are presented as the means ± SEM in (**B**) (*n* = 4); (**C**,**D**) Immunofluorescence staining was conducted at 12 and 24 h after LPS treatment with anti-LC3 antibody (green) and DAPI (blue). Representative images are shown in (**C**) (Scale bar, 20 µm), and quantification of the dot– or ring–shaped LC3 signals (representing autophagosomes) are shown in (**D**) (*n* = 12); (**E**,**F**) ATG7, ATG14, ATG16, and Beclin-1 levels were examined using immunoblotting. Representative immunoblots are presented in (**E**), and quantification results are presented as the means ± SEM in (**F**) (*n* = 3 for ATG7 and ATG14, *n* = 4 for ATG16 and Beclin-1). (**G**,**H**) BV2 cells were pretreated with Bafilomycin A1 (Baf, 50 nM) or chloroquine (CQ, 5 μM) for 2 h and then treated with LPS (500 ng/mL) for 24 h, and level of LC3 was examined using immunoblotting; (**I**,**J**) BV2 cells were treated with LPS (100 ng/mL) for 12 h in the presence or absence of TLR4-IN-C34 (5 and 10 μM), and LC3 levels were examined using immunoblotting. Representative immunoblots are presented in (**H**) and (**I**), and quantification data are presented in (**G**) and (**J**) (*n* = 4). * *p* < 0.05, ** *p* < 0.01 vs. PBS-treated control cells, ^##^
*p* < 0.01 between indicated groups using one–way ANOVA with Student–Newman–Keuls test.

**Figure 2 cells-11-02410-f002:**
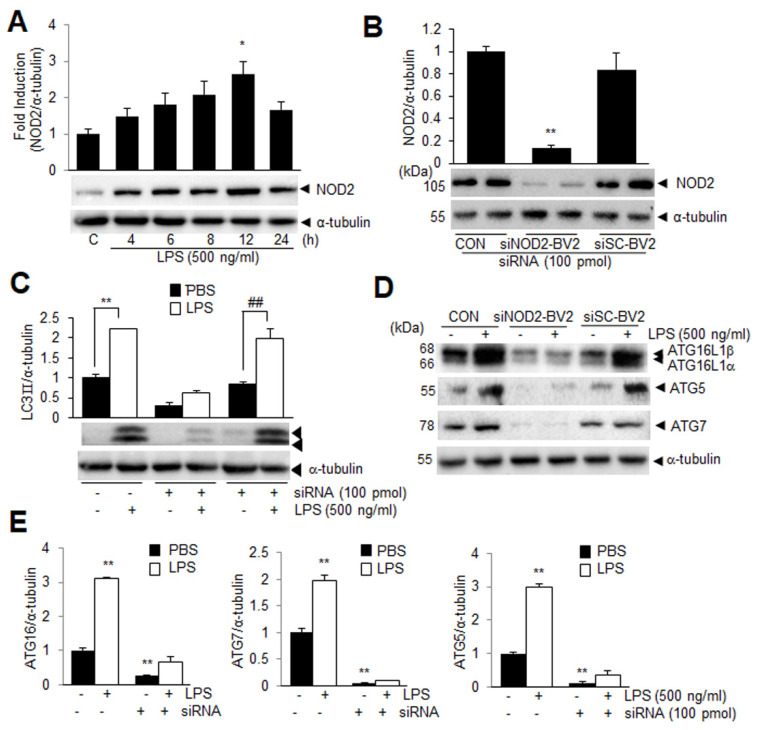
NOD2 deficiency suppresses LPS-induced autophagy in BV2 cells. (**A**) BV2 cells were treated with LPS (500 ng/mL) for 4, 6, 8, 12, and 24 h, and the level of NOD2 was examined using immunoblotting (*n* = 3); (**B**–**E**) NOD2 siRNA or scrambled siRNA were transfected into BV2 cells (siNOD2-BV2 and siSC-BV2 cells, respectively); At 24 h after transfection, the basal NOD2 levels were examined using immunoblot analysis (**B**); At 24 h after transfection, the cells were treated with LPS (500 ng/mL), and the LC3 levels at 12 h post LPS treatment were examined (**C**); Levels of ATG5, ATG7, and ATG16 were examined in BV2, siNOD2-BV2, and siSC-BV2 cells at 12 h after LPS treatment using immunoblotting (**D**,**E**). Representative immunoblots are presented and quantification results are presented as the means ± SEM. * *p* < 0.05, ** *p* < 0.01 vs. PBS-treated control cells, and ^##^
*p* < 0.01 between indicated groups using one–way ANOVA with Student–Newman–Keuls test.

**Figure 3 cells-11-02410-f003:**
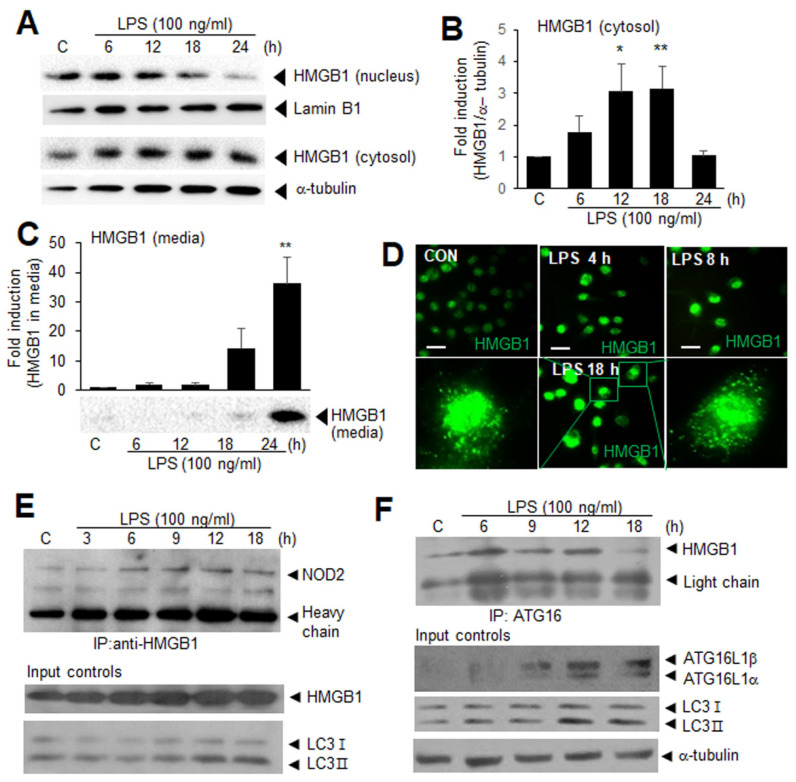
Cytosolic HMGB1 translocated from the nucleus after LPS treatment interacts with NOD2 and ATG16 in BV2 cells. (**A**,**B**) BV2 cells were treated with LPS (100 ng/mL), and the levels of HMGB1 in nucleus and cytosol were examined after 6, 12, 18, and 24 h of LPS treatment using immunoblotting. Representative immunoblots are presented in (**A**) and quantification results of cytosolic is presented as the mean ± SEM (*n* = 4) in (**B**); (**C**) BV2 cells were treated with LPS (100 ng/mL), and the levels of HMGB1 in culture medium were examined after 6, 12, 18, and 24 h of LPS treatment using immunoblotting. Representative immunoblots are presented and quantification result is presented as the mean ± SEM (*n* = 6); (**D**) Immunofluorescence staining was carried out at 4, 8, and 18 h after LPS treatment with the anti-LC3 antibody, and representative images at each time point and high magnification images obtained from 18 h are presented. Scale bar, 50 µm; (**E**,**F**) BV2 cells were treated with LPS (100 ng/mL) for 3, 6, 9, 12, and 18 h, and the interactions between HMGB1-NOD2 (**E**) or HMGB1-ATG16 (**F**) were examined using co-immunoprecipitation. Antibodies used in immunoprecipitation and immunoblotting are presented in each experiment. * *p* < 0.05, ** *p* < 0.01 vs. PBS-treated control cells.

**Figure 4 cells-11-02410-f004:**
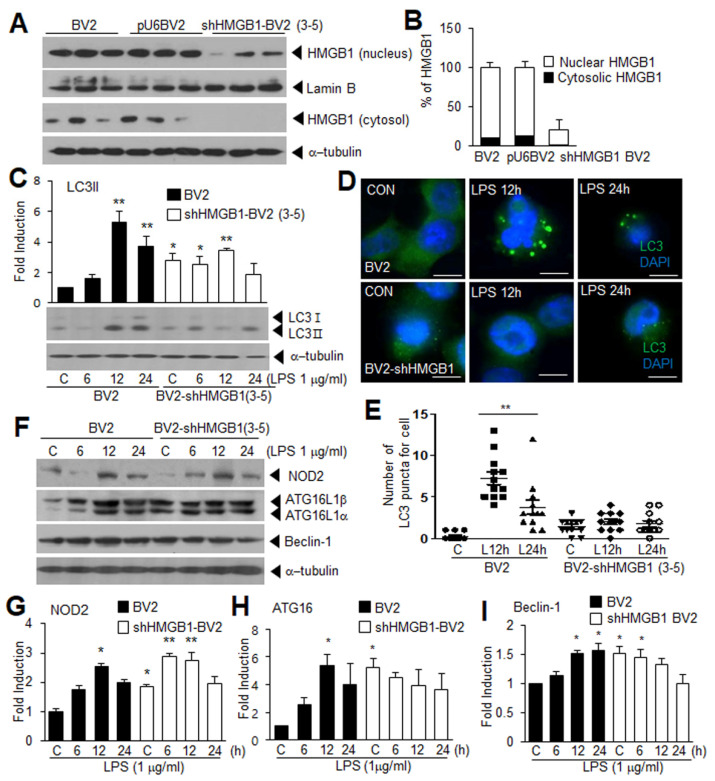
LPS-induced autophagy is significantly suppressed in HMGB1 KD BV2 cells. BV2 cells were transfected with pU6 plasmid expressing HMGB1 shRNA (shHMGB1-BV2) or empty pU6 plasmid (pU6-BV2), and shHMGB1-BV2 (3-5) cell line was selected. (**A**,**B**) Nuclear and cytosolic HMGB1 levels were examined using immunoblot analysis in wild type, pU6-BV2, and shHMGB1-BV2 cells, and representative immunoblot and quantification results are presented (*n* = 3); (**C**,**F**–**I**) Wild-type BV2, pU6-BV2, and shHMGB1-BV2 cells were treated with LPS (1 μg/mL) for 6, 12, and 24 h, and the levels of LC3 (**C**), and NOD2, ATG16, and Beclin-1 (**F**–**I**) were examined using immunoblotting analysis. Representative immunoblots are presented in (**C**) and (**F**), and quantification results are presented as the mean ± SEM (*n* = 4) in (**C**) and (**G**–**I**); (**D**,**E**) Wild-type BV2, pU6-BV2, and shHMGB1-BV2 cells were treated with LPS (1 μg/mL) for 12 and 24 h, and immunofluorescence staining was performed with anti-LC3 antibody (green) and DAPI (blue). Representative images are shown in (**D**), and quantification of the dot- or ring-shaped LC3 signals (representing autophagosomes) are presented as the mean ± SEM (*n* = 12) in (**E**). Scale bar, 20 µm. * *p* < 0.05, ** *p* < 0.01 vs. PBS-treated control cells.

**Figure 5 cells-11-02410-f005:**
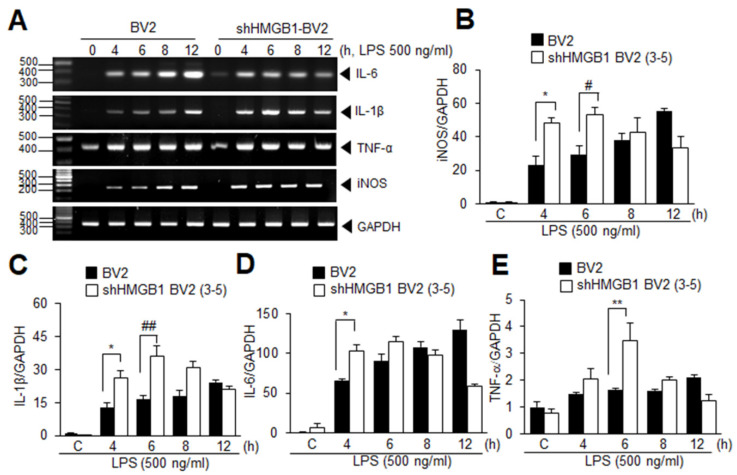
Induction of pro-inflammatory cytokine expression is enhanced in LPS-treated HMGB1 KD BV2 cells. BV2 or shHMGB1-BV2 cells were treated with LPS (500 ng/mL) for 4, 6, 8, and 12 h, and expression of iNOS, IL-1β, IL-6, and TNFα was examined using RT-PCR analysis. Representative results are presented in (**A**) and quantification results are presented as the mean ± SEM in (**B**–**E**). * *p* < 0.05, ** *p* < 0.01 vs. PBS-treated control cells, ^#^
*p* < 0.05, ^##^
*p* < 0.01 between indicated groups using one-way ANOVA with Student–Newman–Keuls test.

**Figure 6 cells-11-02410-f006:**
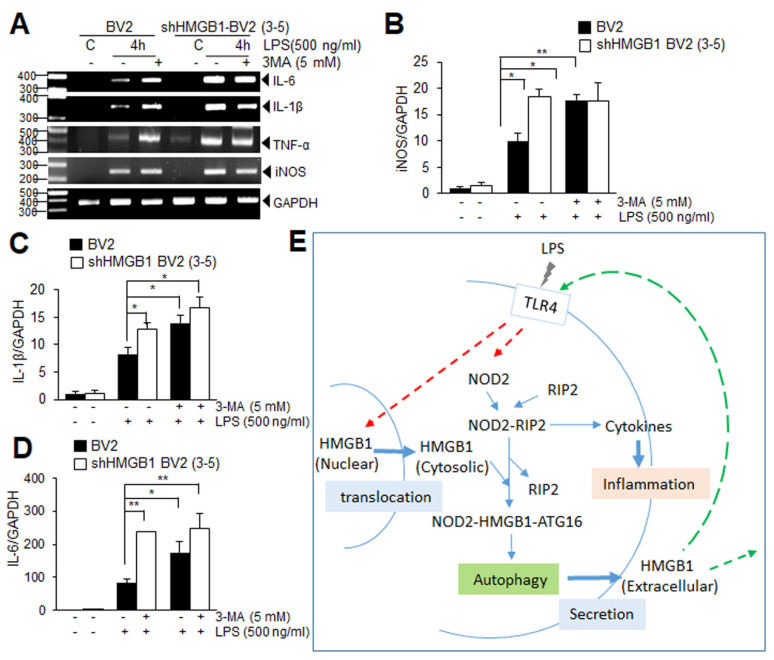
Induction of pro-inflammatory cytokine expression is enhanced in LPS-treated HMGB1 KD BV2 cells. (**A**–**D**) BV2 and shHMGB1-BV2 cells were pretreated with 3-MA (5 mM) for 2 h and then treated with LPS (500 ng/mL) for 4 h, and the expression of iNOS, IL-1β, and IL-6 was examined using RT-PCR. Representative results are presented in (**A**) and quantification results are presented as the mean ± SEM in (**B**–**D**). * *p* < 0.05, ** *p* < 0.01 vs. PBS-treated control cells; (**E**) Schematic representation depicting the modulation of NOD2-mediated autophagy by cytosolic HMGB1 in BV2 cells.

**Table 1 cells-11-02410-t001:** Oligonucleotide primers used in RT-PCR analysis.

Gene (GenBank Accession No.)	Oligonucleotide Primer Sequences	PCR Product Size (bp)	Tm
IL-1β (M98820)	5′-AGC ATC CAG CTT CAA ATC TCA-3′ 5′-CGA GGC ATT TTT GTT GTTCAT-3′	271	54
IL-6 (BC132458)	5′-GGA AAT GAG AAA AGA GTT GTG CAA T-3′ 5′-CCT TAG CCA CTC CTT CTG TGA-3′	370	54
TNF-α (NM012675)	5′-CTC AAA ACT CGA GTG ACA AG-3′ 5′-CTC CGT GAT GTC TAA GTA CT-3′	422	46
iNOS (AY090567)	5′-AAGAACGTGTTCACCATGAGG-3′ 5′-CCAGTAGCTGCCACTCTCATC-3′	363	54
GAPDH (DQ403053)	5′-TCA TTG ACC TCA ACT ACA TGG T-3′ 5′-CTA AGC AGT TGG TGG TGC AG-3′	252	54

## Data Availability

Not applicable.

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
