# Peer review of "Cytosolic HMGB1 Mediates LPS-Induced Autophagy in Microglia by Interacting with NOD2 and Suppresses Its Proinflammatory Function"

_cells, 2022, doi:10.3390/cells11152410_

Round 1

Reviewer 1 Report

The research manuscript of Seung-Woo Kim and co-authors report the therapeutic efficacy of Cytosolic HMGB1 mediates LPS-induced autophagy in microglia by interacting with NOD2 and suppresses its proinflammatory function.

The authors have addressed the potential of HMGB1-induced autophagy in LPS-treated BV2 microglial cell line in mediating the transition between the inflammatory and autophagic function of the nucleotide-binding oligomerization domain containing 2 (NOD2), a cytoplasmic pattern recognition receptor. The underlying mechanisms involve the NOD2-RIP2 interaction, proinflammatory cytokine induction, and NF-B activity were significantly enhanced in BV2 cells after HMGB1 KD, indicating that HMGB1 plays a critical role in the modulation of NOD2 function between proinflammation and proautophagy in microglia. The effects of cell-autonomous proautophagic pathway operated by cytoplasmic HMGB1 in microglia may be beneficial, whereas those from the paracrine proinflammatory pathway executed by extracellularly secreted HMGB1 can be detrimental. This manuscript addresses the overall functional significance of HMGB1-induced autophagy is different depending on its temporal activity.

In general, this study is partially novel and cannot provide potential evidence for the cytosolic HMGB1 mediates LPS-induced autophagy in microglia by interacting with NOD2 and suppresses its proinflammatory function. Therefore, although there is not much research article so far describing about the HMGB1 mediates LPS-induced autophagy in microglia. There are certain weaknesses in the experimental design of the molecular mechanism the author needs to look into it, to make the manuscript publishable form.

Major and Minor concerns:

1.     In Abstract the authors need to be careful in using abbreviations like DAMP and LPS. They can be used in full words when using first time.

2.     In introduction, in final paragraph the authors need to specify their study aims clearly which can accomplish their hypothesis.

3.     In materials and methods, some of the protocols does not show the full details of the experiments, I recommend the authors to include this reference for the above-mentioned protocol, DOI: 10.1016/j.redox.2022.102280 DOI: 10.1016/j.phymed.2021.153648, and DOI: 10.1016/j.phymed.2021.153887.

4.     In figure 1A, the authors need to add blot for SQSTM1 for autophagy induction.

5.     In figure 1, authors need to show the autophagy flux experiment to confirm the autophagy induction, autophagy flux experiment can be done using CQ or Bafilomycin.

6.     In figure 1, the authors need to include autolysosome experiment to show autophagosome and lysosome fusion using Tf-LC3 cells or plasmid and validate the autophagy induction.

7.     The authors need to include the flow cytometry assay to show the transition of microglial cells from proinflammatory stage to anti-inflammatory stage.

8.     The authors can refer these papers (2022)  https://doi.org/10.1016/j.redox.2022.102280,

https://doi.org/10.1038/s41401-022-00871-0 ,  https://doi.org/10.1016/j.apsb.2022.01.017 for the above-mentioned experiments which are very important to prove the autophagy induction.  

9.     The above corrections should be carried out in the figures and the requested experiments.

10.  A careful English check and grammatical errors need to be resolved by the authors.

So, this manuscript precedes about the potential evidence for the cytosolic HMGB1 mediates LPS-induced autophagy in microglia by interacting with NOD2 and suppresses its proinflammatory function. The underlying mechanisms involve the NOD2-RIP2 interaction, proinflammatory cytokine induction, and NF-B activity were significantly enhanced in BV2 cells after HMGB1 KD, indicating that HMGB1 plays a critical role in the modulation of NOD2 function between proinflammation and proautophagy in microglia, which makes this work noble and creates scientific interest for the readers. However, the available research information seems to be insufficient for being accepted in current form. Taking together to all this issue I recommend major revision to the manuscript and in current form the manuscript does not fit for publication in the Cells.

Author Response

Major and Minor concerns:

  1. In Abstract the authors need to be careful in using abbreviations like DAMP and LPS. They can be used in full words when using first time.

Response: By following reviewer’s comment, we included full words for abbreviations in the revised manuscript. In addition, we rewrote abstracts to make it more concise.

  1. In introduction, in final paragraph the authors need to specify their study aims clearly which can accomplish their hypothesis.

Response: By following reviewer’s comment, we rewrote Introduction section in the revised manuscript. We made it concise and clearly mentioned specific aims of this manuscript.

  1. In materials and methods, some of the protocols does not show the full details of the experiments, I recommend the authors to include this reference for the above-mentioned protocol, DOI: 10.1016/j.redox.2022.102280 DOI: 10.1016/j.phymed.2021.153648, and DOI: 10.1016/j.phymed.2021.153887.

Response: By following reviewers comment, we rewrote Materials and Methods section in the revised manuscript. We addressed full details of some experiments, including catalog numbers and dilution factors of all antibodies and concentrations of drugs used. We included an additional reference in the revised manuscript in relation with those correction.

  1. In figure 1A, the authors need to add blot for SQSTM1 for autophagy induction.

Response: By following reviewer’s suggestion, we carried out immunoblot analysis using anti- SQSTM1/p62 antibody. Results showed a gradual decrease of SQSTM1/p62, indicating autophagy induction in LPS-treated BV2 cells. We inserted this new data in Figure 1A and included related descriptions in Materials and Methods and Results sections of the revised manuscript. 

  1. In figure 1, authors need to show the autophagy flux experiment to confirm the autophagy induction, autophagy flux experiment can be done using CQ or Bafilomycin.

Response: We agreed on reviewer’s point. We demonstrated a significant accumulation of LC3 after treating CQ or Bafilomycin which supporting autophagy flux in LPS-treated BV2 cells. We inserted this new set of data in Figure 1G and H and included related descriptions in Materials and Methods and Results sections of the revised manuscript. 

  1. In figure 1, the authors need to include autolysosome experiment to show autophagosome and lysosome fusion using Tf-LC3 cells or plasmid and validate the autophagy induction.

Response: Due to the time limitation of obtaining Tf-LC3 cells or plasmid and conduction the experiment, we cannot carried out the experiment. Please find the evidence in Fig. 1 of the revised manuscript supporting the LPS-induced autophagy induction in BV2 cells including the ones added by reviewer’s suggestion.

  1. The authors need to include the flow cytometry assay to show the transition of microglial cells from proinflammatory stage to anti-inflammatory stage.

Response: We agree on reviewer’s point. Interrelationship between autophagy induction and M1/M2 transition of macrophages and microglia is an important and a big issue. I am also interested in this topic. However, I am afraid to say that we cannot pursue this issue and present a clear answer in this manuscript. Instead, we addressed this issue and discussed about it in the Discussion section of the revised manuscript.

  1. The authors can refer these papers (2022)  https://doi.org/10.1016/j.redox.2022.102280,

https://doi.org/10.1038/s41401-022-00871-0 ,  https://doi.org/10.1016/j.apsb.2022.01.017 for the above-mentioned experiments which are very important to prove the autophagy induction.  

Response: By following reviewer’s suggestion, we included a reference (https://doi.org/10.1016/j.redox.2022.102280) in the Materials and Method section of the revised manuscript.

  1. The above corrections should be carried out in the figures and the requested experiments.

Response: By following reviewer’s suggestion, we tried our best to improve our manuscript by carrying out requested experiments. We described all the changes in the “Response to reviewer “ Section and in the revised manuscript.

  1. A careful English check and grammatical errors need to be resolved by the authors.

Response: We proofread the whole manuscript with great care and corrected mistakes in the revised manuscript.

Reviewer 2 Report

The manuscript “Cytosolic HMGB1 mediates LPS-induced autophagy in microglia by interacting with NOD2 and suppresses its proinflammatory function” by Kim and colleagues discusses pro-inflammatory and pro-autophagic role of NOD2 and its modulation by HMGB1. The manuscript is potentially interesting to explore the novel role of NOD2 and HMGB1 but it lacks a deeper discussion about the achieved results with chaotic presentation especially with the figure legends and figures. In my opinion, the introduction is incomplete, written chaotically, and requires a thorough reconstruction. Following major points that need to addressed before publication.

1.     Abstract is too wordy and I suggest to short it down. Explicitly discuss the theme and outcome of the manuscript.

2.     Introduction should be rewritten mostly chaotic and looks like review of literature on the second paragraph. Be specific what the paper is describing and the hypothesis. Only the last paragraph of the introduction deals with the study.

3.     Mention catalog numbers for all the antibodies used in this study as well the clone used for better reproducibility in the methods section. Also, the dilutions used, only for few it has been mentioned. Indicate the LPS and 3-MA concentrations used in the study in the respective methods section.

4.     Provide the primers used for RT-PCR, since in the manuscript it has been mentioned as Table 1. But nowhere it is presented. 

5.     Entire manuscript is chaotic with legends and figures. To mention a few, Fig 1D doesn’t has scale bar instead it’s on 1C. 1G in legend reads LPS as 100 ng/ml but in actual fig it has been presented as 1 mg/ml and the quantification axis reads 1 μg/ml which indeed confusing to believe the obtained results. 1G significance was not aligned on the bars and what is the comparison between 2nd and rest of the bars on the right indicates? Is it * or # since in legend it reads # but in figure it has presented as *? In 1C what does the insert indicates? Mention the exact number of n in 1F.

6.      What was the rationale in increasing the LPS concentration in NOD2 study (Section 3.2)?

7.     Again, in the Fig 2 the LPS concentration confuses the readers is it 100 or 500 ng/ml or 1 μg/ml. Be explicit in mentioning the concentrations. In fig 2C instead of mentioning LPS- and LPS+, use PBS and LPS to avoid confusion. LPS concentration usage is not consistent thought out the manuscript.

8.     I would suggest to quantify the HMGB1 using ELISA in Fig3A.

9.     Entire figures and its respective legends need to be taken for errors.

10.  English proofreading is important to mitigate grammatical errors and several minor typos are also to be fixed.

11.  I would be happy to recommend this paper for publication if the aforementioned major issues were clarified and recommended suggestions were taken care of.

Author Response

Reviewer 2

  1. Abstract is too wordy and I suggest to short it down. Explicitly discuss the theme and outcome of the manuscript.

Response: By following reviewer’s comment, we rewrote abstracts to make it more concise.

  1. Introduction should be rewritten mostly chaotic and looks like review of literature on the second paragraph. Be specific what the paper is describing and the hypothesis. Only the last paragraph of the introduction deals with the study.

Response: By following reviewer’s comment, we rewrote Introduction section in the revised manuscript. We made it concise and clearly mentioned specific aims of this manuscript.

  1. Mention catalog numbers for all the antibodies used in this study as well the clone used for better reproducibility in the methods section. Also, the dilutions used, only for few it has been mentioned. Indicate the LPS and 3-MA concentrations used in the study in the respective methods section.

Response: In the revised manuscript, we included catalog numbers and dilution factors for all the antibodies used and indicated concentrations of chemicals used in each experiment.

  1. Provide the primers used for RT-PCR, since in the manuscript it has been mentioned as Table 1. But nowhere it is presented. 

Response: In the initial manuscript, we included it in the supplementary Table 1. In the revised manuscript, we included Table 1 in the Materials and Method section.

  1. Entire manuscript is chaotic with legends and figures. To mention a few, Fig 1D doesn’t has scale bar instead it’s on 1C. 1G in legend reads LPS as 100 ng/ml but in actual fig it has been presented as 1 mg/ml and the quantification axis reads 1 μg/ml which indeed confusing to believe the obtained results. 1G significance was not aligned on the bars and what is the comparison between 2ndand rest of the bars on the right indicates? Is it * or # since in legend it reads # but in figure it has presented as *? In 1C what does the insert indicates? Mention the exact number of n in 1F.

Response: We apology for our careless mistakes in Fig. 1. We corrected all of the pointed errors in the revised manuscript and also checked other part of the manuscript carefully. 

  1. What was the rationale in increasing the LPS concentration in NOD2 study (Section 3.2)?

Response: We understand reviewer’s concern. Initially, we carried out all our experiment for autophagy induction at 1 mg/ml LPS. After we found that autophagy was induced at lower concentration of LPS, for example, 100 ng/ml or 500 ng/ml, we used those lower concentration. We replaced all the data in Figure 1 with the ones obtained with 1 mg/ml LPS and additional experiments suggested by reviewer 1 was also carried out at 1 mg/ml LPS.

  1. Again, in the Fig 2 the LPS concentration confuses the readers is it 100 or 500 ng/ml or 1 μg/ml. Be explicit in mentioning the concentrations. In fig 2C instead of mentioning LPS- and LPS+, use PBS and LPS to avoid confusion. LPS concentration usage is not consistent thought out the manuscript.

Response: In Fig. 2, 500 ng/ml of LPS was used for all experiments. We corrected all the mistakes in the legend of Fig. 2. We replaced “–LPS” with PBS in the index by following reviewer’s comment.

  1. I would suggest to quantify the HMGB1 using ELISA in Fig3A.

Response: Due to a time limitation to conduct experiments after purchasing ELISA kit, we repeated immunoblotting experiment for culture media for two more times to reduce error bar and present better quantification data. We replaced Fig. 3C with new ones and rewrote related part in the revised manuscript.

  1. Entire figures and its respective legends need to be taken for errors.

Response: We proofread the whole manuscript with great care and corrected mistakes in the revised manuscript.

  1. English proofreading is important to mitigate grammatical errors and several minor typos are also to be fixed.

Response: We proofread the whole manuscript with great care and corrected mistakes in the revised manuscript.

  1. I would be happy to recommend this paper for publication if the aforementioned major issues were clarified and recommended suggestions were taken care of.

Round 2

Reviewer 1 Report

Most of my previous comments has been answered and the manuscript has been significantly improved. The research article is well written and have discussed the points pertaining their novelty and creates scientific interest for the readers.

The available research information seems to be sufficient and advised for publication. In current form I recommend the manuscript can be published in Cells.

Reviewer 2 Report

The authors have satisfactorily addressed the concerns raised in the initial review. Still several typos are found, kindly fix it during proof reading.